# Concentration and size distribution of black carbon over the ablation area of Potanin glacier: Enrichment ability of surface weathering granular ice of water-insoluble particles with snow/ice melting

Sayako Ueda<sup>1</sup>, Akiko Sakai<sup>1</sup>, Sho Ohata<sup>2</sup>, Purevdagva Khalzan<sup>3</sup>, Sumito Matoba<sup>4</sup>, Ken Kondo<sup>1</sup> and Hitoshi Matsui<sup>1</sup>

<sup>1</sup>Graduate School of Environmental Studies, Nagoya University, Furo-cho, Chikusa-ku, Nagoya 464-8601, Japan <sup>2</sup>Institute for Space–Earth Environmental Research, Nagoya University, Nagoya, 464-8601, Japan <sup>3</sup>Information & Research Institute of Meteorology Hydrology & Environment, Ulaanbaatar 15160, Mongolia

Abstract. Light-absorbing particles on surface ice in ablation areas can accelerate glacier melting and shrinkage. A Single Soot Particle Photometer was used to measure black carbon (BC) mass concentrations ( $M_{BC}$ ) in the ablation area of Potanin Glacier, Mongolia during summer. Surface-ice  $M_{BC}$  values (42–555 ng g<sup>-1</sup>) greatly exceeded those of surface snow (5–22 ng g<sup>-1</sup>), snow and rain (2–6 ng g<sup>-1</sup>), and surface melt water (2–11 ng g<sup>-1</sup>). Vertical profiles of  $M_{BC}$  revealed high surface-layer concentrations, suggesting impurities trapped in the granular ice: the particularly low-density layer on the surface of the weathering crust. In the ablation area,  $M_{BC}$  values of granular ice decreased with lower elevation: 134–601 ng g<sup>-1</sup> at 3317 m site and 8–96 ng g<sup>-1</sup> at 3078 m site. The fraction of residual surface BC to BC contained in lost water over a year, R was calculated using the yearly BC deposition flux and water ablation weight  $A_w$ . Average R values were 0.17 and 0.011, respectively, at 3317 m and 3078 m.  $A_w$  were 246 g w.e. cm<sup>-2</sup> and 325 g w.e. cm<sup>-2</sup>, suggesting that the granular ice retains BC particles best in the upstream ablation area, showing concomitantly less capability with increasing ablation. Enriched BC on the ablation area surface comprises recent BC deposits and BC from the glacier's lower layer after rising during decades or more. Those BC emissions and deposits can therefore affect both future and present ablation area melting processes.

25

<sup>&</sup>lt;sup>4</sup>Institute of Low Temperature Science, Hokkaido University, Sapporo 060-0819, Japan

<sup>10</sup> Correspondence to: Sayako Ueda (ueda.sayako.u2@f.mail.nagoya-u.ac.jp)

Graphical abstract.

# 1 Introduction

Black carbon (BC) is emitted into the atmosphere through combustion activities, such as burning fossil fuels and biomass. The light absorption properties of BC can influence the Earth's radiation budget strongly through atmospheric processes and positive feedback on snow and ice albedo, consequently exacerbating the melting of snow and ice after deposition on surfaces (e.g. Bond et al., 2013; Hansen and Nazarenko, 2004; Mori et al., 2019; Ramanathan and Carmichael, 2008; Skiles et al., 2018). Because of global warming, the area covered by mountain glaciers has been shrinking in recent years (Hugonnet et al., 2021; Khalzan et al., 2022; The GlaMBIE Team, 2025). Over decades, glaciers flow from high-altitude accumulation areas, where

temperatures are below freezing, to low-altitude ablation areas (Cuffey and Paterson, 2010). At ablation areas, surface snow and ice melt and flow out through water channels, either into or over the glacier. Understanding the relation between snowmelt and light-absorbing particles in ablation areas is necessary for comprehension of the glacier shrinkage process.
Most glacier studies have used thermal—optical carbon analysis (e.g. Li et al., 2017, 2018; Zhang et al., 2017, 2020). By contrast, although used only rarely for glacier studies (Kaspari et al., 2011; Konya et al., 2021; Marquetto et al., 2020), nebulizer and

Single Soot Particle Photometer (SP2) systems have been applied for BC measurements supporting ice core and snow research (Bisiaux et al., 2012ab; Goto-Azuma et al., 2024, 2025; Kaspari et al., 2011; McConnell et al., 2007; Mori et al., 2016, 2019, 2020; Osmont et al., 2018; Shinha et al., 2018; Sigle et al., 2018; Sterle et al., 2013). Unlike the thermal–optical method, which measures particles accumulated in a filter sample, the SP2, an aerosol measuring instrument, counts individual BC particles and their mass individually. This count provides a size–number/ mass distribution of BC, which is useful for elucidating particle dynamics and albedo. The SP2 in wide-range mode, which replaces the detector used to measure the aerosol shell with

one that measures large BC particles, can measure BC, including large particle sizes that the standard SP2 is unable to detect (Mori et al., 2016).

Several earlier glacier reports have described concentrations of black carbon (BC) and other impurities, such as mineral dust and organic matter (OM). Thermal—optical measurements of BC have often been used to assess BC levels around the Tibetan Plateau, where concentrations higher than 1000 ng/g have been reported (Li et al., 2017, 2018; Ming et al., 2016; Qu et al., 2014; Xu et al., 2012; Zhang et al., 2017, 2020). Based on measurements of impurities on glaciers, surface albedo has also been estimated, suggesting that BC can contribute to reducing albedo in some places (Zhang et al., 2020). The mass concentration of BC in aged snow tends to be higher than in fresh new snow: it consequently increases with lower elevation over the accumulation glacial area (Li et al., 2017, 2018; Zhang et al., 2017, 2020), suggesting BC enrichment over snow surfaces. The enrichment with impurities is attributed to sublimation, evaporation, and snow melting (Aoki et al., 2014; Doherty et al., 2013; Kuchiki et al., 2014; Ming et al., 2016; Qu et al., 2014; Xu et al., 2012). Nevertheless, quantitative discussions of enrichment and the principles of enrichment in melting warrant support based on further research. Particularly, observational studies addressing ablation areas affected by intense melting are few. Consequently, factors controlling BC concentrations contributing to melting over the glacier surface in the ablation area remain unclear.

For this study, we collected snow and ice samples from the Potanin Glacier in Mongolia, located in the East Asian high mountain range, with particular examination of the ablation area. BC mass concentrations and size distributions were measured using a wide-range SP2. Observations were conducted during the summer (July–August) of 2022 for vertical profiles, in 2023 for different snow and ice samples (precipitation, snow, ice, and water over the glacier surface), and in 2024 for altitude differences of surface granular ice (the low-density surface layer of the weathering crust, which can be readily sampled using a shovel). Based on the observed BC concentrations, one can infer the enrichment process of BC particles over the surface ice of the glacier ablation area. Furthermore, this study demonstrates and evaluates a method for quantitative estimation of the residual ability of BC particles in surface granular ice at each location.

https://doi.org/10.5194/egusphere-2025-5301 Preprint. Discussion started: 12 November 2025 © Author(s) 2025. CC BY 4.0 License.

# 2 Field Observations and Laboratory Methods

# 2.1 Observation site and sampling

Potanin Glacier is located in the Tavan Bogd region, which lies along the border separating Mongolia, Russia, and China. The Tavan Bogd region, which is designated as a national park, is protected from residential development, making it a particularly popular tourist destination. Potanin Glacier, which flows from the summit of Mt. Khuiten (4,374 m above sea level (a.s.l.)), is the highest peak among the Altai Mountains situated eastward. The glacier extends 10.4 km along a valley and terminates at 2,907 m a.s.l. (Figure 1). The glacier area is estimated as 24.7 km². Its mass balance has been observed using stakes at 50–100 m elevation intervals on the Potanin (14 stakes), as described by Khalzan et al. (2022, 2025). For this study, surface snow, and ice or water samples were collected from around four stakes (st5r, 4c, 3c, and 2c), which were located at the midpoint of each elevation, and the accumulation area point (ACC) shown in Figure 1. Details of each stakeholder are presented in Table 1. The glacier flow velocity was measured by application of feature tracking (Sakakibara and Sugiyama, 2014, 2020) to image pairs of Landsat 9 images acquired on 28 July 2022 and 31 July 2023. We used an automatic image matching scheme known as the orientation correlation method in a frequency domain (Heid and Kääb, 2012; Sakakibara and Sugiyama, 2020). Horizontal biases in the image pair were corrected by minimizing displacement vectors over ice-free areas. Vectors with a low signal-tonoise ratio (<30) and those deviating by more than 30° or 100 m a<sup>-1</sup> from the median vector within 3 × 3 neighboring pixels were excluded from analyses. Errors in the velocity measurement were estimated as ±1.7 m a<sup>-1</sup> based on results obtained for stable ice-free areas.

Figure 1: Satellite photograph of Potanin Glacier and locations of stakes.

Table 1: Elevations for each stake, glacier flow velocity, and atmospheric temperature and ice temperature of ACC, st5r, st4c, st3c, and st2c sites

|      | Elevation Glacier flow velocity |                         | Atmospheric temperature [°C] | Ice temperature [°C] |        |         |
|------|---------------------------------|-------------------------|------------------------------|----------------------|--------|---------|
|      |                                 | [m year <sup>-1</sup> ] | Ave. [MinMax.]               | -20 cm               | -50 cm | -100 cm |
| ACC  | 3757                            | _                       |                              |                      |        |         |
| C. F | 3317                            | 59                      | 4.2 [-0.5 – 12.1]            | 0.03                 | -0.6   | -1.7    |
| St5r |                                 |                         | (2023/7/12 - 8/11)           | (2023/7/2)           |        |         |
| St4c | 3232                            | 27                      |                              |                      |        |         |
| St3c | 3150                            | 1.5                     |                              |                      |        |         |
| St2c | 3078                            | 1.0                     | 6.3 [1.3–11.8]               | 0.03                 | -0.8   | -2.0    |
| 3120 |                                 |                         | (2023/7/12 - 8/11)           | (2023/7/3)           |        |         |

https://doi.org/10.5194/egusphere-2025-5301 Preprint. Discussion started: 12 November 2025 © Author(s) 2025. CC BY 4.0 License.

Figure 2 presents examples of the sampling site. Meltwater flowed through channels on the glacier's surface in the ablation area. On 12 July 2023, the glacier was covered in white aged snow, which by 8 August had melted away and exposed weathering crust covered with granular ice layer at the surface after experiencing some melting and refreezing. Weathering crust over st5r, 4c, 3c, and 2c were dark compared to the snow-covered surface. And colored impurities were clearly concentrated in the granular ice on the surface of the weathering crust layer. Snow, ice and water samples were collected according to the following classification: surface snow, surface ice (i.e. the surface granular ice), surface water (i.e. meltwater) and precipitation (i.e. snow and rain). These samples are listed in Tables S1-S4 in Supplementary Materials. Using a stainless steel scoop, snow and ice samples were collected in dust-free plastic bags. After melting the snow and ice in the plastic bags at base camp, the resulting liquid was stored in 30 ml or 50 ml glass bottles (SV30 and SV50; Nichiden Rika Glass Co., Ltd.) for BC measurements. Some samples were collected in 10 ml PET vials (JST-R/N10 and JST-R/N30; Nikko Hansen & Co., Ltd.) for TEM analysis. Surface water was collected directly in glass bottles. Precipitation samples were collected in dust-free plastic bags over a basin at the base camp. In 2023, we collected samples particularly addressing different snow types. In 2024, we collected samples to obtain statistical data on surface granular ice. In 2022, 4.2 m and 3.0 m ice core samples were collected at st2c on 31 August and at st3 on 1 September. Based on our earlier evaluation of storage methods (Ueda et al., 2025), BC mass concentrations were obtainable from liquid samples stored at ~4 °C that were comparable to freshly melted samples. However, refreezing can lead to a considerably large (>20% on average) loss. Therefore, the samples were kept liquid and were transported to Nagoya University by refrigerated delivery, where they were stored at 4 °C until measurement.

Figure 2: Examples of sampling site views, 2023, and the BC mass concentrations.

# 2.2 BC measurement using a nebulizer-SP2 system

The mass and number size distributions of BC in liquid samples were measured using a system consisting of a pneumatic nebulizer (Marin-5; Cetac Technologies Inc., USA) and the SP2 (Mori et al., 2016, 2019, 2020, 2021; Sinha et al., 2018). Details of the measurement system used for this study were described by Ueda et al. (2025). Liquid samples were injected into the nebulizer at a constant flow rate of 0.18 mL min<sup>-1</sup> to be aerosolized. The aerosolized particles were subsequently introduced into the SP2 at a flow rate of 0.12 L min<sup>-1</sup>. The SP2 detects individual BC particles from their incandescence signals and BCfree particles from their light-scattering signals in the carrier gas. A standard SP2 measures the mass of each BC particle with mass equivalent diameter ( $D_{BC}$ ) within the 70-850 nm range by assuming a BC particle density of 1.8 g cm<sup>-3</sup> (Moteki and Kondo, 2010), whereas the SP2 used for this study measures masses of BC particles within the 70–3000 nm range by expanding the upper limit of the detected incandescence signal in the standard SP2 (Mori et al., 2016). The number concentration ( $N_{BC}$ ) and mass concentration ( $M_{BC}$ ) of BC in water for the 70–3000 nm diameter range were derived from the flow rates of the liquid sample and a carrier gas for the nebulizer, the nebulizer efficiency, and the measured size distribution of BC in the carrier gas. Before measurement using the nebulizer-SP2 system, liquid samples in glass vials were sonicated for 10 min to minimize the 125 loss of BC particles attached to the vial wall and to mitigate the possible change of the size distribution by the coagulation of BC particles in the samples. From many glacier samples in this study, turbidity and large dust particles were found. To prevent clogging of the nebulizer-SP2 system, after settling of large dust particles by standing for 10 min after sonication for all samples, suspended liquid samples in vial middles were dispensed into 10 ml PET vials (JST-R/N10; Nikko Hansen & Co., Ltd.); then they were measured using the nebulizer-SP2 system. At the dispensing, samples were diluted 3-10 times using Milli-Q water 130 according to their turbidity.

# 2.3 Mineral dust and water-insoluble organic matter

All samples used to measure mineral dust and water-insoluble organic matter were put into plastic bags, including all surface granular ice layer samples with impurities. After taking samples, we measured the sampled square area (length and width) and the sampling depth, i.e. the thickness of surface granular ice layer in the surface part of the weathering crust, at four points of the sides of the sample square. The plastic bag was brought to the base camp. All ice was melted during one day, along with deposited impurities. All of the deposited impurities were put into 30 ml plastic bottles with formalin (1% of samples). After taking samples to the laboratory, the samples were dried at 60°C for 24 h in pre-weighed crucibles. Then we ascertained the dried impurity amounts (mineral dust and organic matter) of samples. After removing the organic matter from dried samples by combustion at 500°C for 3 h in an electric furnace, we found the amount of organic matter from the difference in weight between the dried and combusted samples. Those methods were based on those reported by Takeuchi and Li (2008). Then, the amounts of mineral and organic matter per area were obtainable using the area measured when the samples were collected.

# 2.4 TEM analyses

Particles nebulized by Marin-5 from liquid sample in 10 ml PET vials were collected using a three-stage cascade impactor (50% cutoff aerodynamic diameters of the two stages were 1 and 0.3 μm at a 0.5 L min<sup>-1</sup> flow rate) on the TEM grid. Particles were collected on carbon-coated nitrocellulose (collodion) films over a Cu TEM grid (H7, 400 mesh, Reference Grids; Graticules Optics Ltd., UK). A water dialysis technique (Mossop, 1963; Okada, 1983; Ueda et al., 2011ab) was used to remove water-soluble materials from the samples. The TEM grid with particle samples was floated on an ultrapure water drop (approximately 0.3 ml) on a Petri dish at about 25 °C for 3 h with the collection side upward. To avoid charging up and to obtain the height information of individual particles on the collection surface, particles were coated with a Pt/Pd alloy at a shadowing angle of 26.6° (arctan 0.5) before being micrographed. After water dialysis, residue materials on the TEM grid were photographed using TEM (JEM 2100-plus; JEOL, Japan) with 200 kV accelerating voltage. Elemental mapping was done using EDS (EX-24200M1G2T; JEOL Ltd., Japan) with TEM in scanning TEM (STEM) mode and 200 kV accelerating voltage. Elemental mapping data were obtained and analyzed using software (NSS 3; Thermo Fisher Scientific Inc., USA).

### 155 2.5 Global model simulation

2 reanalysis.

BC deposition over the observation site was estimated using global model simulations with the Community Atmosphere Model ver. 5 coupled with the Aerosol Two-dimensional bin module for foRmation and Aging Simulations (CAM-ATRAS) (Matsui, 2017; Matsui and Mahowald, 2017; Matsui et al., 2018). This model incorporates a comprehensive set of aerosol and atmospheric processes, including emissions, gas-phase chemistry, condensation and evaporation of inorganic and organic species, coagulation, nucleation, aerosol activation and in-cloud processing, dry and wet deposition, and interactions with radiation and clouds. Aerosols are represented using 12 dry diameter bins of 1–10,000 nm. For particles of 40–1,250 nm, eight BC mixing states are considered: BC-free, pure BC, and internally mixed BC-containing particles of six types. Simulations were conducted for 2013–2023, with analyses particularly addressing 2014–2023. The model was run at a horizontal resolution of 1.9° × 2.5°, with 30 vertical layers extending up to approximately 40 km. Anthropogenic emissions were based on the Community Emissions Data System 2021 inventory (O'Rourke et al., 2021) and were assumed to be constant at 2019 levels for 2020–2023 because the inventory is available only through 2019. Sea surface temperatures and the sea ice extent were prescribed according to an earlier report by Hurrell et al. (2008). Horizontal winds and temperature in the free

Against ground-based, aircraft, and satellite observations, CAM-ATRAS has been evaluated extensively. It has shown good agreement with observed BC concentrations at the surface, in vertical profiles, and on a global scale (e.g., Liu and Matsui, 2021a, 2021b; Matsui and Mahowald, 2017; Matsui et al., 2022).

troposphere (above 800 hPa) were nudged toward the Modern-Era Retrospective analysis for Research and Applications ver.

### 3 Results and Discussion

# 3.1 BC and other impurities over the ablation area

# 175 3.1.1 BC concentrations in snow, surface ice, water, and precipitation, 2023

Figure 3 presents the mass concentrations of BC in water ( $M_{BC}$ ) found for surface snow, surface ice, surface water, snow, and rain samples collected in 2023. Some  $M_{BC}$  values are shown in Figure 2. The  $M_{BC}$  values of precipitation (i.e., snow and rain) were 2–6 ng g<sup>-1</sup>. The  $M_{BC}$  values of surface snow were 5–22 ng g<sup>-1</sup>: slightly higher than those of precipitation. The values of surface ice were especially high, 42–555 ng g<sup>-1</sup>, with a maximum value found for st5r. Surface water originates from the melt of surface snow and ice. However, the  $M_{BC}$  values for surface water were 2–11 ng g<sup>-1</sup>, which are considerably less than those found for surface ice. The value at st5r in July, when surface snow covered the area, was low: 3 ng g<sup>-1</sup>. However, when surface granular ice (surface ice) exposed in August, the value at st5r was highest among surface water, 11 ng g<sup>-1</sup>, but it was sufficiently lower than that for surface ice, 555 ng g<sup>-1</sup>.

Figure 3: Mass concentrations of BC in water (M<sub>BC</sub>) for surface snow, surface ice, surface water, snow, and rain samples collected during 30 June – 12 August in 2023.

The concentrations of BC at the surface of East Asian glaciers reportedly have widely variant values: from less than 10 ng g<sup>-1</sup> to larger than 1000 ng g<sup>-1</sup> (Li et al., 2017, 2018; Ming et al., 2016; Qu et al., 2014; Xu et al., 2012; Zhang et al., 2017, 2020). In fact, BC concentrations higher than 3000 ng g<sup>-1</sup> were observed for granular ice in the Xiao Dongkemadi glacier of the

https://doi.org/10.5194/egusphere-2025-5301 Preprint. Discussion started: 12 November 2025 © Author(s) 2025. CC BY 4.0 License.

Tibetan Plateau (Li et al., 2017). In general, the concentrations of impurities (e.g. BC, mineral dust, and OM) tended to be higher in aged snow and surface granular ice than in fresh snow over the accumulation glacial area (Li et al., 2017, 2018; Ming et al., 2016; Qu et al., 2014; Zhang et al., 2017, 2020). Sublimation and evaporation can condense the impurities which are present in surface snow (Aoki et al., 2014). Furthermore, earlier studies have found that snow impurities tend to concentrate at the surface during melting (Doherty et al., 2013; Ming et al., 2016; Qu et al., 2014; Xu et al., 2012). The present study also showed that  $M_{BC}$  tended to be low for fresher snow and high for aged surface ice. In addition, the  $M_{BC}$  of surface water was lower than that of surface ice. These results suggest that most BC particles were retained in the ice rather than flowing out with the water.

### 3.1.2 BC, mineral dust, and organic matter in surface ice for altitudes, 2024

Figure 4 portrays box plots of the mass concentrations of BC, mineral dust, and organic matter in water (respectively, *M*<sub>BC</sub>, *M*<sub>Dust</sub>, and *M*<sub>OM</sub>) for surface ice samples collected in 2024. Surface ice conditions were aged snow ice for ACC place, and surface granular ice for stakes 5r, 4c, 3c, and 2c. The *M*<sub>BC</sub>, *M*<sub>Dust</sub>, and *M*<sub>OM</sub> of surface granular ice in ablation (st5r, 4c, 3c, and 2c) vary widely, respectively ranging between 8–600 ng g<sup>-1</sup>, 0.9–40 mg g<sup>-1</sup>, and 3–1483 μg g<sup>-1</sup>. However, those values for all stakes were higher for dark ice, which mainly covered the area, than for white ice, a minor area. As in the 2023 trend, the *M*<sub>BC</sub> values of surface ice for dark ice and all samples tended to be highest at st5r (134–601 ng g<sup>-1</sup>); they decreased by one order with lower altitudes (8–96 ng g<sup>-1</sup> at st2c). However, such an order change of mass concentration with altitude was not found for mineral dust or for organic matter.

The amount of organic matter can be influenced by biological activity on ice. Mineral dust in the ablation area can also be supplied from rocks in the glacier and the bottom of the glacier. Because there are no specific local factors for increase and decrease in BC, the amount is controlled mainly by wet deposition, enrichment and dilution in a glacier, and outflow. Several reports have described trends of increasing impurity concentrations with low elevation in the accumulation areas of glaciers in Central Asia (Li et al., 2017, 2018; Zhang et al., 2020), which can be attributed to snow aging processes, including sublimation and evaporation, as well as snowmelt amplification. The BC mass concentration in surface ice found from our study was also the highest at the ablation area near the accumulation area (st5r). However, they decreased in areas of lower elevation, which is the inverse of trends reported for accumulation areas in earlier studies. In the ablation area, water is lost from the surface mainly by out flow of the passing water channel rather than by sublimation or evaporation. The flowing water can run off some BC. Therefore, enrichment effects of BC with snowmelt might decrease in the downstream region of the ablation area.

Figure 4: Bbox plots of the mass concentration of BC, mineral dust, and organic matter in water ( $M_{\rm BC}$ ,  $M_{\rm Dust}$ , and  $M_{\rm OM}$ ) for surface ice samples collected during 22 July – 2 August in 2024. The left boundary of the box shows the 25th percentile. The line within the box represents the median. The right boundary of the box stands for the 75th percentile. The whiskers left and right of the box respectively show the 90th and 10th percentiles.

3.1.3 BC size distributions and particle mixing states

225

Figure 5 presents normalized mass–size distributions of BC in the samples, 2023 and 2024, used in Figs. 2 and 3. Averages, minimum and maximum values of  $M_{\rm BC}$ , and lognormal fitting results are presented in Table 2 for 2023 and in Table 3 for 2024. Most of the size distributions have peak diameters of 200–400 nm. The size distribution for snow and surface snow samples in 2023 often had sharp shapes around the peak. The modal geometric standard deviations,  $\sigma_{\rm gm}$  are, respectively, 1.76–1.81

and 1.82–2.36. The  $\sigma_{\rm gm}$  values are approximately equal to the values reported for atmospheric aerosols (1.63–1.76 over the north polar region, as measured by aircraft, Ohata et al., 2021), surface snow (1.66–2.21 at Arctic regions by Mori et al., 2019), and snowfall (1.5–2.1 at Ny-Ålesund by Mori et al., 2021). However, the mass–size distributions for surface ice tended to have broad size distributions ( $\sigma_{\rm gm}$  is 2.06–2.43). The distributions of water samples were broad for most samples ( $\sigma_{\rm gm}$  is 1.77–2.36). For surface snow samples in 2024, including aged snow ice at ACC, most of the distributions were also wide (average of  $\sigma_{\rm gm}$  for dark ice at each stake is 2.41–3.00). Although  $M_{\rm BC}$  had trends depending on altitudes as explained in Sec. 3.1.2, no clear trend was found for the shape of size distributions.

Figure 5: Normalized mass-size distributions of BC in the samples: 2023 and 2024.

Table 2: Average [Minimum-Maximum] values of BC mass concentrations and parameters of mass-size distribution for samples collected in 2023

|             | $M_{ m BC} \left[ { m ng \ g^{	ext{-}1}}  ight]$ |             |            | MAD*E 1      | D ** [ ]                     | **               |
|-------------|--------------------------------------------------|-------------|------------|--------------|------------------------------|------------------|
|             | 70–850 nm                                        | 850–3000 nm | total      | $MAD^*$ [nm] | ${D_{\mathrm{gm}}}^{**}[nm]$ | $\sigma_{ m gm}$ |
| Rain        | 2.8                                              | 0.8         | 3.6        | 190          | 317                          | 2.11             |
|             | [1.5–4]                                          | [0.2–2]     | [1.7–6.1]  | [156–228]    | [194–443]                    | [1.62–2.41]      |
| Snow        | 2.3                                              | 0.3         | 2.6        | 185          | 254                          | 1.78             |
|             | [1.8–2.9]                                        | [0.3–0.3]   | [2.1–3.2]  | [180–191]    | [249–260]                    | [1.76–1.81]      |
| Surface     | 8.3                                              | 1.6         | 9.9        | 176          | 251                          | 1.99             |
| snow        | [4.3–18.3]                                       | [0.3–4.4]   | [5.4–22.7] | [161–190]    | [215–270]                    | [1.82–2.36]      |
| Surface ice | 223                                              | 28.9        | 252        | 169          | 252                          | 2.21             |
|             | [38.7–499]                                       | [3.2–55.8]  | [42–555]   | [164–172]    | [229–270]                    | [2.06–2.43]      |
| Surface     | 4.8                                              | 0.7         | 5.5        | 171          | 253                          | 2.12             |
| water       | [1.7–9.3]                                        | [0–1.6]     | [1.8–10.9] | [152–191]    | [201–300]                    | [1.77–2.36]      |

<sup>\*</sup> Mass averaged diameter (MAD) was estimated as  $(6M_{\rm BC}/\pi\rho_{\rm BC}N_{\rm BC})^{1/3}$ .  $\rho_{\rm BC}$  and  $N_{\rm BC}$  respectively represent , the BC mass density (=1.8 g cm<sup>-3</sup>) and number concentration of BC.

Table 3: Average [Minimum–Maximum] values of BC mass concentrations and parameters of mass–size distribution for samples collected in 2024

|      | $M_{ m BC}$ [ng g <sup>-1</sup> ] |                  |                |                  | $MAD^*[nm]$      | D **[mm]                            | ${\sigma_{ m gm}}^{**}$ |
|------|-----------------------------------|------------------|----------------|------------------|------------------|-------------------------------------|-------------------------|
|      |                                   | 70–850 nm        | 850–3000 nm    | total            | MAD [IIII]       | $D_{\mathrm{gm}}^{**}[\mathrm{nm}]$ | $O_{\rm gm}$            |
| ACC  |                                   | 179<br>[33–325]  | 66<br>[15–118] | 245<br>[48–443]  | 190<br>[188–192] | 333<br>[316–350]                    | 2.5<br>[2.48–2.53]      |
| St5r | dark                              | 229<br>[109–476] | 59<br>[24–124] | 288<br>[133–600] | 176<br>[161–187] | 294<br>[238–326]                    | 2.41<br>[2.23–2.58]     |
|      | white                             | 91<br>[11–317]   | 22<br>[5–61]   | 113<br>[17–379]  | 185<br>[178–192] | 351<br>[306–388]                    | 2.72<br>[2.28–3.21]     |
| St4c | dark                              | 61<br>[28–103]   | 13<br>[2–29]   | 74<br>[31–133]   | 160<br>[150–169] | 224<br>[182–284]                    | 2.4<br>[1.92–2.96]      |
|      | white                             | 28<br>[13–36]    | 5<br>[3–11]    | 34<br>[16–45]    | 154<br>[148–161] | 206<br>[182–234]                    | 2.7<br>[2.26–3.53]      |
| St3c | dark                              | 47<br>[23–101]   | 9<br>[0–23]    | 56<br>[24–124]   | 154<br>[124–175] | 209<br>[139–295]                    | 2.82<br>[1.75–4.26]     |
|      | white                             | 23<br>[13–42]    | 8<br>[2–14]    | 31<br>[20–56]    | 174<br>[165–178] | 302<br>[243–429]                    | 3<br>[2.41–4.55]        |
| St2c | dark                              | 26<br>[4–60]     | 12<br>[1–36]   | 38<br>[7–96]     | 191<br>[162–229] | 324<br>[194–470]                    | 2.49<br>[1.6–3.18]      |
|      | white                             | 8<br>[6–11]      | 3<br>[2–4]     | 11<br>[8–16]     | 182<br>[165–200] | 349<br>[251–503]                    | 2.78<br>[2.61–2.96]     |

<sup>\*</sup> Mass averaged diameter (MAD) was estimated as  $(6M_{\rm BC}/\pi\rho_{\rm BC}N_{\rm BC})^{1/3}$ .  $\rho_{\rm BC}$  and  $N_{\rm BC}$  respectively represent , the BC mass density (=1.8 g cm<sup>-3</sup>) and number concentration of BC.

\*\* Modal geometric mean diameter ( $D_{\rm gm}$ ) and the modal geometric standard deviation ( $\sigma_{\rm gm}$ ) by mono-modal lognormal fitting of mass-size

<sup>\*\*</sup> Modal geometric mean diameter ( $D_{gm}$ ) and the modal geometric standard deviation ( $\sigma_{gm}$ ) by mono-modal lognormal fitting of mass–size distribution for BC less than 1000 nm.

<sup>\*\*</sup> Modal geometric mean diameter ( $D_{gm}$ ) and the modal geometric standard deviation ( $\sigma_{gm}$ ) by mono-modal lognormal fitting of mass—siz distribution for BC less than 1000 nm.

https://doi.org/10.5194/egusphere-2025-5301 Preprint. Discussion started: 12 November 2025 © Author(s) 2025. CC BY 4.0 License.

Major factors of change in BC size distributions for ice and snow samples after deposition are coagulation of BC particles and size-dependent out flow with melted water, depending on the particle size. However, the BC size distributions of surface water and surface ice were similar; the dependence of BC flow out was unclear. Related to the former reason, Figure 6 shows STEM images and elemental maps of water-insoluble particles collected at the smallest particle stage for a snow sample and two surface ice samples. Aggregations of globules of less than 50 nm diameter, which are characteristic of soot particles (Janzen, 1980; Murr and Soto, 2005; Pósfai et al., 2004; Ueda et al., 2022), were found in a snow sample (Fig. 6a). However, we were unable to find such bare soot particles from surface ice samples because many other particles were dominant for the surface 265 ice samples. For example, insoluble particles in surface ice samples at st5r were mainly aggregated particles, composed primarily of C, O, and Si (Fig. 6b). Although their shape is similar to that of soot, the individual spherules in the aggregation had a larger diameter (>100 nm) and were O-rich compared to soot, indicating organics. In surface ice at st2c, most of the insoluble particles were angular particles composed mainly of Si, O, and Al, regarded as aluminosilicates (Fig. 6c). The minor mass concentration of BC compared to those of mineral dust and organic matter corresponds to results of mass concentrations, as shown in section 3.1.2. However, in general, submicrometer atmospheric aerosols are rarely composed only of such organics or mineral dust in internal mixed states (Adachi et al., 2022; Li et al., 2016; Ohata et al., 2018; Pósfai et al., 2004; Ueda et al., 2016, 2022, 2023). Therefore, such submicrometer insoluble particles were probably sediments formed after deposition or broken pieces, originating from larger dust and organic matter. Mass concentrations of BC, mineral dust, and OM in surface ice were markedly higher than those in precipitation samples. The coexistence of numerous particles, including sediments, 275 might accelerate the coagulation and aggregation of BC and other particles, consequently moderating the sharp peak in the BC size distribution.

Figure 6: STEM images and elemental maps of water-insoluble particles collected at the smallest particle stage: (a) soot and aluminosilicate in a snow sample at base camp (23BC#04); (b) fine OM particles in surface ice samples at st5r (23BC#25), and fine aluminosilicate particles in surface ice sample at st2c (23BC#30).

# 3.2 BC vertical profile

### 3.2.1 Snow under the lower accumulation area, 2023

Figure 7 presents snow types and BC mass concentrations of the snow pit from the surface to 24 cm depth at the ACC site in 2023. A water-bearing layer and an ice layer were found, respectively, at depths of 17 cm and 20 cm. The upper layer consisted of melt forms and rounded-grain layers, which probably accumulated after a snowmelt event. The BC mass concentration was lower (3.7 ng g<sup>-1</sup>) for the sample containing the water-bearing layer than the upper snow layers, but higher (84.0 ng g<sup>-1</sup>) for the sample containing the ice layer. This structure implies the separation of BC from the upper layer with snowmelt and enrichment at the ice layer.

Figure 7: Vertical profile of BC and snow types for snow pit in ACC site, 2023.

# 3.2.2 Core samples at the lower ablation area, 2022

Figure 8 presents a vertical profile of BC, OM, and mineral dust for core samples collected at st2c, 31 August 2022, and at 295 st3c, 1 September 2022. For st2c, a high-concentration layer of BC, dust, and organic matter (respectively, 64 ng g<sup>-1</sup>, 4.4 mg g<sup>-1</sup>, and 0.2 mg g<sup>-1</sup>) was found at depths of 15–20.5 cm. However, those concentrations from just below the layer to below were low (respectively, <16 ng g<sup>-1</sup>, <2.0 mg g<sup>-1</sup>, and <0.01 mg g<sup>-1</sup>). Similar trends were observed for st3c: a high concentration

layer at depths of 0–15 cm and a low concentration below this layer. The vertical profiles suggested that the event of high impure-particle deposition, equivalent to the BC concentration of the surface ice, was rare in this observation site. Therefore, the high concentration of BC near the surface could be attributed mainly to melting processes at the surface of snow and ice, rather than to the wet and dry deposition of atmospheric particles.

Figure 8: Vertical profile of BC, organic matter, and mineral dust for ice core samples drilled at st2c on 31 August 2022 (a) and at st3c on 1 September 2022. Cases of no data because of storage vial cracking are blanked.

# 3.2.3 Surface wet snow and the underlying weathering granular ice

Weathering granular ice layers with high concentrations of BC were located within 8 cm below the surface (Tables S4 and S5). However, the layers were sometimes covered by wet snow. Figure 9 portrays the number–size and mass–size distributions of BC for snow/ granular ice samples collected at st2c, 3 July 2023. The BC concentrations of the granular ice for most BC sizes were higher than those of the upper wet snow layer, which was 2–3 cm thick. Both the BC mass size distribution of upper wet snow and lower granular ice had a similar broad shape.

Figure 9: Number-size and mass-size distribution of BC for surface wet snow and the underlying weathering granular ice samples collected at st2c, 3 July 2023.

To understand such surface structure and BC enrichment with snow melting, we conducted a simple test for snow melting using a snow block in st2c 2024. After the original snow in a plastic bag was exposed to the sun for 3.5 h, the melted water and residual snow were collected separately. Detailed results are presented in Figure S1 of Supplementary Information. The BC concentration of the melted water (15.4 ng g<sup>-1</sup>) was higher than that of the residual snow (4.5 ng g<sup>-1</sup>). Light absorption by BC particles can explain this result: snow around BC particles preferentially melted and fell below, consequently being stored at the bottom of the plastic bag in this experiment. Over the glacier surface at the ablation area, the down-flowing water can partly flow out to water channels, be kept partly in the ice, and be refrozen. The high concentration of BC in granular ice might therefore be formed through the retention and refreezing of flowed-down, BC-enriched water.

# 3.3 Enrichment and outflow of BC over the ablation area

# 325 3.3.1 Inference of the BC enrichment process

As described in Sections 3.1.1 and 3.2.1, the BC concentration in surface ice that had experienced melting and refreezing (i.e. weathering granular ice and ice layers) was significantly higher than in snow and wet snow. This higher concentration suggests that BC is enriched by the melting of snow and ice. As shown in Section 3.1.2, the BC concentration in surface ice was highest in St5R (the upstream part of the ablation area) and decreased towards the downstream area. This result suggests that the balance between retained and lost BC is different for different elevations. Several reports have described enrichment with melting (Doherty et al., 2013; Ming et al., 2016; Qu et al., 2014; Xu et al., 2012), but the mechanism remains unclear. As explained in Section 3.2.2, light-absorbing particles absorb shortwave radiation and consequently sink preferentially as they melt the surrounding snow. However, layers with a high BC concentration were near the surface. The BC concentration of surface flowing water was one or two orders of magnitude lower than the surrounding surface ice. This low concentration suggests that many light-absorbing particles are trapped in the ice below the surface and that water with a low BC concentration is preferentially flowing out. Based on inference from earlier studies and this observation, insoluble particles remain for several possible reasons. 1. Light-absorbing particles that settle preferentially refreeze first in the lower layer, where the ice temperature is low. This refreezing creates a hard ice plate that fixes particles in place. 2. When melting, the granular ice in the lower layer acts as a filter, allowing only water to flow out through the channels. 3. At night, the concentrated layer of black carbon (BC) in the lower layer freezes. However, the diluted surface layer of BC melts because of heat conduction from the atmosphere, flowing out and leaving behind the solid lower layer. Several factors can influence the ability of ice to retain BC: the structure and density of the weathering granular ice, the melting rate and

# 3.3.2 Remaining BC mass in surface granular ice

As shown in Figure 8, the concentration of impurities in the surface granular ice was markedly higher than that below the surface. Therefore, impurities are regarded as enriched in surface -granular ice. The surface granular ice usually had 1–8 cm thickness (Tables S3 and S4). Figure 10 shows the weight of surface granular ice per unit area for the dark ice covering most of the surface for st5r, 4c, 3c, and 2c in 2023 and 2024, as well as the total and size-segregated (70–850 nm and 850–3000 nm) black carbon (BC) mass in the surface granular ice in 2024. The masses of surface granular ice ( $m_{WC}$ ) and BC mass in the surface granular ice ( $m_{SurfBC}$ ) were, respectively, 0.2–4.4 g cm<sup>-2</sup> and 5.7–896 ng cm<sup>-2</sup>.

frequency, as well as the elevation and temperature of the atmosphere, surface, and layers.

Figure 10: Relation of elevations with weight of surface granular ice per unit area (a) and total BC (b) in 2023 and 2024, and size-segregated BC mass (c and d) in the surface granular ice for st5r, 4c, 3c, and 2c in 2024. The dark ice samples, which covered most of the surface, were used, whereas white ice was excluded because of a minor area.

Annual BC deposition flux calculated using a global model CAM-ATRAS are listed in Tables S5 in Supplementary Materials. Based on calculations from a global model (CAM-ATRAS), annual deposition in this region over the past 10 years was estimated as 896–1,494 ng cm<sup>-2</sup> per year. The recharge rate of Potanin, calculated using a mass balance model based on the

mass balance obtained from the stake observations (2003–2018), is 621 mm yr<sup>-1</sup> in terms of precipitation (Khalzan et al., 2022).

By multiplying this quantity by the average BC concentration of snow samples collected in 2023, which is 2.7 ng g<sup>-1</sup>, the roughly calculated annual flux is 1675 ng cm<sup>-2</sup> yr<sup>-1</sup>, a value similar to the BC deposition predicted by the model, thereby supporting the validity of the model results. In addition, based on measurements of the stakes' surface height, the yearly snow/ice loss ( $A_w$ , water ablation weight) at st5r, 4c, 3c, and 2c are, respectively, 246 g cm<sup>-2</sup> (=cm water equivalent), 274 g cm<sup>-2</sup>, 264 g cm<sup>-2</sup>, and 325 g cm<sup>-2</sup> per year on a three-year average. Considering that BC mass concentration under BC-enrichment layer ( $M_{underBC}$ ) is 4.4 ng g<sup>-1</sup> on average in ice cores from 15 cm to 3 m deep, the BC weight in ablated snow under the surface ( $A_{BC}$ ) is estimated roughly as 1,080–1,429 ng cm<sup>-2</sup> as  $M_{underBC} \times A_w$ . Both the annual deposition of BC and  $A_{BC}$  were higher than, but of the same order as, the highest  $m_{surfBC}$  (896 ng cm<sup>-2</sup>) in in st5r.

Although  $m_{WC}$  changed less with elevation, the  $m_{surfBC}$  tended to be exponentially smaller at lower altitudes. For size-segregated BC, the exponential index slope was slightly larger for particles smaller than 850 nm. However, the difference was not great. These results suggest that the retention of BC particles in the surface granular ice decreases concomitantly with decreasing altitude, irrespective of particle size.

# 3.3.3 Estimation of BC remaining fraction on surface ice with meltwater out flow

As explained in Section 3.3.2, the values of BC in surface granular ice were smaller than the yearly deposition of BC and the amount of BC in ablated water, but they might be comparable. As discussed in Section 3.3.1, various processes related to altitude and temperature can affect the concentration of BC particles in surface ice. For this study, we calculate the surface granular ice's ability to retain BC particles at each stake in the ablation region, based on observed BC concentrations and snow ablation rates. We estimated the surface remaining fraction of BC to BC contained in lost water over a year (R: annual remaining fraction) as an indicator of the surface granular ice's ability. The mass of BC remaining on the ablation surface in a given year ( $m_{\text{surfBC}}(t)$ ) can be expressed in terms of the previous year's mass ( $m_{\text{surfBC}}(t-1)$ ) as shown below.

$$m_{surfBC}(t) = (m_{surfBC}(t-1) + F_{BC} + A_{BC}) \times R$$

$$A_{BC} = M_{underBC} \times A_{w}$$

$$0 \le R \le 1$$

$$(1)$$

Therein,  $F_{\rm BC}$  represents the deposition flux of BC from air for a year. Also, as  $F_{\rm BC}$ , 1228 ng cm<sup>-2</sup> year<sup>-1</sup> of averaged values of 2021–2023 was used, as estimated using CAM-ATRAS. Although  $M_{\rm underBC}$  is strictly variable, averaged value were used (=4.4 ng g<sup>-1</sup>). Equation (1) is expanded as presented below.

$$m_{surfBC}(t) = (m_{surfBC}(t=0)) \times R^{t} + (F_{BC} + A_{BC}) \times (R + R^{2} + R^{3} + \dots + R^{t})$$

$$= (m_{surfBC}(t=0)) \times R^{t} + (F_{BC} + A_{BC}) \times \frac{R(1-R^{t})}{(1-R)}$$

The value of  $m_{\text{surfBC}}(t)$  converges to the following.

$$m_{surfBC} \simeq (F_{BC} + A_{BC}) \times \frac{R}{(1 - R)}$$
 (3)

This status of  $m_{\text{surfBC}}(t)$  and  $m_{\text{surfBC}}(t-1)$  is mostly the same, independent of t. The value of  $m_{\text{surfBC}}(t)$  converges more quickly if R is smaller. As presented in Fig. 10, the  $m_{\text{surfBC}}$  values of 2024 were the values of 2023 for st2c, 4c, and 5r, which can be regarded as  $m_{\text{surfBC}}(t) \simeq m_{\text{surfBC}}(t-1)$ , an almost converged state, at these sites. Therefore, R is developed using (3) to the expression presented below.

$$R \simeq \frac{m_{surfBC}}{m_{surfBC} + F_{BC} + A_{BC}} \tag{4}$$

Strictly speaking, *R* can depend on *t* because they differ according to elevation. However, the glacier flow velocities at st5r, 4c, 3c and 2c are, respectively, 59, 27, 1.5, and 1.0 m year<sup>-1</sup> (Table 1). The horizontal distances between st5r and 4c, 4c and 3c, and 3c and 2c are, respectively, 1.4, 1.4, and 1.3 km. Even at the rapid upstream flow from st5r to st4c, it takes longer than two decades. The movement and elevation changes of stakes can be regarded as less within a year. Here, we considered *R* change less on a few-year scale in this case and used equation (4). In this case,  $F_{BC} + A_{BC}$  corresponds to BC, which runs off in a year per unit of the area.

0 Table 4: BC mass in the surface granular ice (m<sub>surfBC</sub>) in 2024, water ablation weight (Aw), BC in annual ablated snow (A<sub>BC</sub>), and annual remaining fraction of BC (R) for each stake

|      |                  |              | $A_{ m BC}^*$ [ng BC cm <sup>-2</sup> year <sup>-1</sup> ] | R*                  |
|------|------------------|--------------|------------------------------------------------------------|---------------------|
|      | Avg. [Min.–Max.] | 3 yr average |                                                            | Avg. [Min.–Max.]    |
| St5r | 517 [234–896]    | 246±25       | 1080                                                       | 0.17 [0.09–0.28]    |
| St4c | 92 [50–129]      | 274±38       | 1206                                                       | 0.036 [0.02–0.05]   |
| St3c | 96 [20–192]      | 264±27       | 1162                                                       | 0.038 [0.008–0.074] |
| St2c | 30 [12–70]       | 325±11       | 1429                                                       | 0.011 [0.004–0.026] |

<sup>\*</sup> Values estimated assuming  $M_{\rm underBC}$ =4.4 ng g<sup>-1</sup> and  $F_{BC}$ =1228 ng cm<sup>-2</sup> year<sup>-1</sup>

The estimated R values and the values used for estimation are presented in Table 4. The average values of R are 0.17, 0.036, 0.038, and 0.011, respectively, for st5r, 4c, 3c, and 2c, suggesting a decreasing trend in the ability of snow to retain BC particles downstream. Despite an increase in  $A_w$  of only approximately 30% from St5r to St2c, the calculated R value for St2c was one order of magnitude lower than that for St5r. By contrast, the R values were almost identical at St4 and St3, despite their

different altitudes, because both the  $A_w$  and  $m_{surfBC}$  values were almost identical at these two locations. The  $m_{surfBC}$ , residual BC in surface granular ice, might also depend on the  $A_w$ , relating to factors such as the structure of granular ice and the water flow pressure. The rate of water loss might have a stronger effect on the particle retention capacity of granular ice than elevation. As explained in section 3.3.1, the enhancement of BC with the snow/ice melting process is related to multiple processes. An increase in  $A_w$  with low elevation can affect multiple factors, including the reach length to a water channel and the structure of the weathering granular ice. Such an increase might consequently decrease the particle filtered efficiency of granular ice. Doherty et al. (2013), based on surface and subsurface snow BC concentrations observed at Barrow, estimated the scavenging fraction for snow melting as 10-30% (i.e. a remaining fraction of 0.7-0.9). In this study, the R value for the accumulated lost water over the year is lower than the remaining fraction (0.7-0.9) because it represents the remaining fraction to a more extended period (one year), incorporating the effects of multiple factors including the melting event frequency and the surface ice structure, as well as the period during which the granular ice is covered by snow accumulation in winter. Although the remaining fractions at different locations differed considerably on an annual scale, assuming roughly three months of snowmelt, the surface remaining fraction of BC for a day ( $R_{\rm day}$ ) can be estimated as  $R^{1/(90 \, {\rm days})}$ : 0.98, 0.96, 0.96, and 0.95, respectively, for st5r, 4c, 3c, and 2c, which indicates that most BC is retained in the surface ice on a daily basis. That finding corresponds to the observation that the BC concentration in surface water is nearly two orders of magnitude lower than that in surface ice.

### 4 Summary and Conclusions

We investigated the black carbon (BC) mass concentration ( $M_{BC}$ ) in snow and ice samples from the ablation area of Potanin Glacier in Mongolia during the summers of 2022, 2023, and 2024. Of the various snow and ice types, surface snow had slightly higher  $M_{BC}$  values (5–22 ng g<sup>-1</sup>) than precipitation (2–6 ng g<sup>-1</sup>), although these were much lower than those of surface ice (42–555 ng g<sup>-1</sup>). Additionally, the  $M_{BC}$  values for surface water (2–11 ng g<sup>-1</sup>) originating from the melting of surface ice were low. Vertical profiles of BC showed a high concentration of BC in the surface layer and a low concentration below it. These results suggest that impurities were enriched in the surface ice layer, probably because of the presence of particles in the weathering granular ice and because of the flow of particle-free water out of the ablation area. The BC mass–size distribution of surface snow, ice, and water typically peaks in the range of 200–400 nm and has a broader shape than that of fresh snow and rain. These results, together with the coexistence of dust and organic matter as observed using TEM, suggest coagulation with enrichment of impurity particles.

The  $M_{BC}$  values over the ablation area in 2024 exhibited a clear decreasing trend with elevation, with the highest (134–601 ng g<sup>-1</sup>) at the st5r (3317 m) site and one order of magnitude lower (8–96 ng g<sup>-1</sup>) at the st2c (3078 m) site. In the ablation area, water is lost from the surface mainly by flow out of the passing water channel rather than by sublimation and evaporation. The enrichment effect of BC with snowmelt might decrease in the downstream region of the ablation area. By contrast, for mineral dust, no change in the order of mass concentration with altitude was observed. However, vertical profiles indicated an enrichment trend of organic matter, dust and BC at the glacier surface. Factors affecting the increase and decrease in BC among

https://doi.org/10.5194/egusphere-2025-5301 Preprint. Discussion started: 12 November 2025

© Author(s) 2025. CC BY 4.0 License.

impurities are limited primarily to deposition and the balance of dilution and enrichment within glaciers. This characteristic of limitation differs from that of organic matter, which can be affected by biological activity in the ice, and mineral dust, which can originate from the glacier itself and its base. For the case in which the BC concentration is regarded as less affected by local emissions, as in this study, the BC concentration can be an indicator of the enrichment process which occurs with melting. Considering BC vertical profiles and simple experiments involving a snow block in a plastic bag, light-absorbing particles absorb shortwave radiation and consequently sink preferentially as they melt the surrounding snow. However, BC-enriched layers were observed near the surface, suggesting that light-absorbing particles were trapped in the surface granular ice and that they rarely flowed out. Various factors can influence the ability of the granular ice to retain BC, including the weathering granular ice structure, melting conditions, and the atmosphere and ice temperatures. For this study, the residual surface rate of BC to BC contained in lost water over a year (*R*: the annual remaining fraction) in the ablation region was calculated using yearly snow/ice loss (*A*<sub>w</sub>: the water ablation weight). The average *R* values were 0.17 at st5r, *A*<sub>w</sub>=246 g cm<sup>-2</sup>, and 0.011 at st2c, *A*<sub>w</sub>=325 g cm<sup>-2</sup>. These findings suggest that the ability to retain BC particles is greater where there is less snow/ice ablation and suggest that it tends to decrease downstream.

For this study, we estimated the enriched BC concentration through glacier melting and evaluated the ability of surface granular ice to retain BC residue in the ablation area during summer. Unlike the accumulation area, where surface snow is converted into fresh snow, the surface ice in the ablation area consisted of melted and refrozen snow that had migrated from the lower layer. Consequently, enriched BC particles on the surface of the ablation area contain recently deposited BC, as well as comparable quantities of BC that have accumulated and migrated over decades or centuries. It is noteworthy that black carbon deposition from the atmosphere, including the effects of human activities, can influence current and future glacier melting.

Author contributions. AS designed the study. SU wrote the paper. AS, PK, and SM contributed to the observation in Potanin glacier. KK calculated glacier flow velocity. SU, SO and AS analysed samples. HM conducted numerical model simulations using CAM-ATRAS.

Competing interests. The authors state that there are no competing interests.

Data availability. Observation data for BC, dust and OM and simulation results by CAM-ATRAS in this study is publicly available through the Supplementary Materials. Satellite data were obtained from Data Workspace (https://dataspace.copernicus.eu/analyse/data-workspace). Other data will be provided upon request.

# 470 Acknowledgments

We acknowledge Ms. Azusa Kishi of Nagoya University for BC analysis. We are indebted to Prof. Michihiro Mochida and Prof. Kazuo Osada of Nagoya University for their support of the experimentation environment. We also extend our gratitude for technical support from the High Voltage Electron Microscope Laboratory of Nagoya University.

#### 475 Financial support.

This study was supported by Science and Technology of Japan and the Japan Society for the Promotion of Science (MEXT/JSPS) KAKENHI Grant nos. 20H00196, 22K18023, 22K18024, 25H00507, 22H03722, 23H00515, 23H00523, 23K18519, 23K24976, and 24H02225, the MEXT Arctic Challenge for Sustainability II (ArCS II; JPMXD1420318865) and phase 3 (ArCS-3; JPMXD1720251001) projects, and by the Environment Research and Technology Development Fund 2–2301 (JPMEERF20232001) of the Environmental Restoration and Conservation Agency, and by National Institute of Polar Research (NIPR) through Special Collaboration Project no. B25-02. This work was achieved partly through the use of SQUID at the D3 Center, Osaka University.

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
