# Peer review of "Concentration and size distribution of black carbon over the ablation area of Potanin glacier: Enrichment ability of surface weathering granular ice of water-insoluble particles with snow/ice melting"

_EGUsphere, 2025_

## Author Comment (AC2)

*Reviewer: 2*

*Comments*

*Ueda et al., 2025 "Concentration and size distribution of black carbon over the ablation area of Potanin glacier: Enrichment ability of surface weathering granular ice of water-insoluble particles with snow/ice melting" presents BC in various materials, snow, ice, and water, which were took from Potanin Glacier. TEM analysis was conducted to investigate fine particle composition of inpurities in snow and ice. They discussed comprehensive observation results of BC deposition, residual, and outflow and estimated BC remaining fration on ice with melt water out flow. This study will contribute to extending understanding of BC dynamics in cryosphere science and more detailed understanding on BC climate effects. The methods and analysis support well the conclusion of this study. There are some parts to be revised.*

Response

We would like to thank you for the many constructive comments, which have helped us to improve the manuscript.

*Major comment*

*1. FBC was obtained by CAM-ATRAS to estimate BC remining fraction. The idea is interesting and challenging. However, CAM-ATRAS is a global model with spatial resolustion of 1.9° × 2.5° and I wonder this resolution is sufficient or not for estimation FBC in cases where significant spatial heterogeneity is anticipated, such as in ablation glacial regions.*

Response

We also think that actual differences in $F_{BC}$ can affect $R$ estimation when meteorological conditions are heterogeneous. However, the surrounding terrain is flat. The ablation area of this glacier has a gentle slope (about 7 degrees). In fact, according to the precipitation parameter (based on accumulation data estimated with pollen content) observed in our earlier study (Khalzan et al., 2022), precipitation between the st2–st5 altitudes was similar (about 11% difference). Considering this similarity and the contribution of $F_{BC}$ to the $R$ calculation formula, it is unlikely that actual $F_{BC}$ would be sufficiently different to alter the conclusion about the altitude difference of $R$. We added a relevant explanation for the stability of precipitation in 3.3.2 and for the terrain in 2.1 to the revised text.

*2. It was difficult to understand the connection between the argument presented at the end of Section 3.1.3 regarding the mixing state of particles and the comparison with atmospheric aerosols. Detailed*

*comments are written in specific comments.*

Response

We revised the explanations in 3.1.3 of the revised text in line with your specific comments and with the other reviewer's comments. Please see related details in response to the specific comments.

*3. Does the explanation of BC enrichment on the snow surface discussed in 3.1.1 and 3.1.2 contradict BC enrichment in melted water discussed in 3.1.3? I think it would be better to organize the discussion again.*

Response

We did not discuss BC enrichment in 3.1.3 before the revision, although we did address the size distribution and coagulation process. We do not think that there was a contradiction. As presented in section 3.1.1, BC enrichment is slight in surface snow (sublimation and evaporation are the main processes), but intense in surface ice (enrichment through snowmelt, detailed in the later section 3.3.1, is the main process). The BC size distributions were also similar and sharp in fresh snow and surface snow, but they exhibited some broadening in surface ice, suggesting a relation to BC enhancement with melting. During revision, the discussion was reorganized into sections 3.1.3 and 3.1.4 in the submitted text.

*Spcific comments*
*L53: I think either 'fresh' or 'new' would be fine.*

Response
We deleted 'new'.

*L69: Add more specific methods for sampling method and sample treatment, such as amount of snow sampled, area of the sampling points, melting techniques, etc. Glacier might have a large spatial discrepancy of BC in snow, and strong heating can decrease measured BC concentration by SP2.*

Response
We added some sample information to the second paragraph of section 2.1 of the revised text. We collected an approximately 10 cm × 10 cm area, as added to the revised text. The weights of the BC

samples were not measured. The surface granular ice depth was added in Tables S3 and S4. In 2024, considering the spatial discrepancy, we collected multiple samples for each stake. The samples were melted under ambient temperature (about 7 °C) at the base camp.

*L75: Since glaciers are flowing, and sampling points are marked by stakes, does this mean the ground position of the sampling points is changing? If so, wouldn't the amount of snow fall and sediment deposited from surrounding weathered rock change over time?*

Response
As you have said, the stakes move. However, because the glacier has a gentle slope (7 degrees) and a slow flow velocity (Table 1), the elevation changes little during a few years. In response to major comment 1, we added that the glacier has a gentle slope in this section of the revised text.

*L90: It is preferable to indicate that a blank field has not been analyzed, such as by labeling it "N.A.," rather than leaving it blank.*

Response
"N.A." was added to Table 1.

*L115: How much air flow rate of the nebulizer?*

Response
The air flow rate was 0.8 L min$^{-1}$. That information was added to the explanation in the revised text.

*L124: I can understand sonication is needed to minimize BC wall loss, but I wonder the sonication is really effective to mitigate the possible change of the BC size? If yes, sonication may also change original BC size distribution. If there are appropriate references, it should be added.*

Response
Because the primary objective is to measure the total amount, the samples were sonicated. Sonication is probably an efficient method for dispersing BC particles that adhere weakly to wall surfaces and to the other particles, including coarse particles that are unable to pass through the nebulizer system. The

sentence was revised as shown below.

Before: "Before measurement using the nebulizer-SP2 system, liquid samples in glass vials were sonicated for 10 min to minimize the loss of BC particles attached to the vial wall and to mitigate the possible change of the size distribution by the coagulation of BC particles in the samples."

After: "Before measurement using the nebulizer-SP2 system, liquid samples in glass vials were sonicated for 10 min to minimize loss of BC that is unable to pass through the nebulizer system, such as BC attached to the vial wall and BC on the coarse particles."

For the sonication size distribution, we decided that this should be considered for each sample feature. Therefore, the description was deleted here. In our earlier test experiments conducted using snow collected from Greenland and Sapporo (0.2–60 ng g$^{-1}$ BC concentration), the concentration recorded immediately after sonication and melting slightly increased, but the particle size distribution changed less (Ueda et al., 2025). However, for samples containing high-concentration impurities, such as the surface ice samples in this study, we think greater care should be taken to account for the possibility of aggregation, as discussed in sections 3.1.3 and 3.1.4 of the revised text.

*L143: Is this a handmade instrument?*

Response
Yes, this is a handmade cascade impactor.

*L196: 'fresh' may be more accurate than 'fresher'. Please check.*

Response
That was revised to be "fresh" in the revised text.

*L214: Compared to what is it 'lower'?*

Response
The sentence was revised as shown below.

Before: "However, they decreased in areas of lower elevation, which is the inverse of trends reported for accumulation areas in earlier studies."

After: "However, they decreased with lower elevation among stakes 5r, 4c, 3c, and 2c, which is the inverse of trends reported for accumulation areas in earlier studies."

*L258: Is there any reference?*

Response
No, we have not found reference material including discussion of the BC size distribution for surface ice samples. Although it was difficult to ascertain the reasons for differences in BC size distributions in snow and ice samples, we revised the paragraph (third paragraph in section 3.1.3) and added a discussion based on our results.

*L270: I could not understand what this sentence is trying to explain. BC is affected post-deposition process, thus comparison with atmospheric BC is meaningless. In addition, while BC sources are remote site from observation area, mineral dust sources are assumed to be local. What are you trying to explain by bringing up atmospheric conditions from entirely different environments shown in these references? Please reconsider.*

Response
As you have commented, we also think that BC in surface ice affected the post-deposition process because the feature in submicrometer water-insoluble particles in the surface ice samples (organic-only or silicates-only states) differed clearly from general water-insoluble materials in aerosol samples. We clarified the distinction with atmospheric insoluble particles to avoid misleading readers into thinking that the particles in the sample are solely of atmospheric origin in revised section 3.1.4.
In the revised version of the text, the references were limited to representative atmospheric states around the Asian atmosphere or to water dialysis. We also added information related to atmospheric aerosol samples to the supplement. Most major aerosol particles are sulfates, with occasional soot-containing particles. However, the submicron water-insoluble matter remaining after water dialysis did not consist predominantly of OM and silicate particles, such as those observed in the surface ice samples. We revised sections 3.1.3 and 3.1.4 of the revised text considering your comments.

*L272: As I pointed above, the source of dust is expected to be neighboring exposed rock area. I think that the mineral dust simply emitted from neighboring areas into the air by saltation and sand blast process, and was deposited on the snow and ice surface (just dry deposition).*

Response

As we have explained in an earlier comment, we also think most of the original dust particles in the glacier originate from the local environment. We agree with that. In addition, silicate, carbonate and other minerals in such dust particles can partially melt and re-form poorly soluble sediment with submicrometer particles such as calcium silicate and calcium carbonate.

*L274: This sentence also fails to clarify what authors expected to explain and its evidence. Why does the presence of other particles promote BC aggregation? If TEM analysis showed advanced BC aggregation or internal mixing with others in the case of samples rich in minerals or organic, this contradicts previous explanations. Furthermore, considering the nebulizer orifice diameter, the droplets formed during atomization should be quite large. Can the agglomeration occurring within droplets really be ignored for these particle rich samples? In any case, I believe this section of the discussion, including this part, requires reorganizing what authors expected to show and their supporting evidence.*

Response

We revised the descriptions and structures of sections 3.1.3 and 3.1.4 to address inconsistencies related to ultrasonic dispersion. Although we sonicated sample vials, some particles, such as strongly connected BCs, might remain connected. The surface ice sample contained high concentrations of impurities: minerals, organic matter, and BC. Therefore, it is thought that particles readily contact one another in melt water before refreezing. In addition, aggregates containing C and Si were found in the TEM sample of the st5r surface ice. Silicates and calcites can be formed as sediment in aqueous solutions and might bond BCs and cover.

We cannot completely rule out that coagulation in the nebulizer might have affected BC high-concentration samples. However, this effect is thought not to be a major factor in the difference in size distributions between snow and ice samples because such highly concentrated samples were measured after dilution by pure water, as explained in section 2.2. In addition, the BC size distribution of surface water with low BC concentration also showed a wide distribution, as shown in Figure 5, suggesting a weaker relationship between the wide distribution and the BC concentration of the samples.